# Variants Affecting the C-Terminal Tail of UNC93B1 Are Not a Common Risk Factor for Systemic Lupus Erythematosus

**DOI:** 10.3390/genes12081268

**Published:** 2021-08-19

**Authors:** Sarah Kiener, Camillo Ribi, Irene Keller, Carlo Chizzolini, Marten Trendelenburg, Uyen Huynh-Do, Johannes von Kempis, Tosso Leeb

**Affiliations:** 1Institute of Genetics, Vetsuisse Faculty, University of Bern, 3012 Bern, Switzerland; sarah.kiener@vetsuisse.unibe.ch; 2Dermfocus, University of Bern, 3001 Bern, Switzerland; 3Division of Immunology and Allergy, Department of Medicine, Lausanne University Hospital (CHUV) and University of Lausanne (UNIL), 1011 Lausanne, Switzerland; camillo.ribi@chuv.ch; 4Interfaculty Bioinformatics Unit, University of Bern, 3012 Bern, Switzerland; irene.keller@dbmr.unibe.ch; 5Department of Pathology and Immunology, School of Medicine, Geneva University, 1211 Geneva, Switzerland; carlo.chizzolini@unige.ch; 6Laboratory for Clinical Immunology, Department of Biomedicine and Division of Internal Medicine, University Hospital of Basel, 4031 Basel, Switzerland; marten.trendelenburg@usb.ch; 7Division of Nephrology and Hypertension, Inselspital, Bern University Hospital, 3010 Bern, Switzerland; Uyen.Huynh-Do@insel.ch; 8Division of Rheumatology, Cantonal Hospital St. Gallen, 9007 St. Gallen, Switzerland; Johannes.VonKempis@kssg.ch

**Keywords:** *Homo sapiens*, immunology, autoimmunity, candidate gene, TLR7 signaling

## Abstract

Systemic lupus erythematosus (SLE) is a heterogeneous multifactorial disease. Upregulated TLR7 signaling is a known risk factor for SLE. Recently, it was shown that specific genetic variants in *UNC93B1* affect the physiological regulation of TLR7 signaling and cause characteristic autoimmune phenotypes with monogenic autosomal recessive inheritance in mutant mice and dogs. We therefore hypothesized that homologous variants in the human *UNC93B1* gene might be responsible for a fraction of human SLE patients. We analyzed 536 patients of the Swiss SLE Cohort Study for the presence of genetic variants affecting the C-terminal tail of UNC93B1. None of the investigated patients carried bi-allelic *UNC93B1* variants that were likely to explain their SLE phenotypes. We conclude that genetic variants affecting the C-terminal tail of UNC93B1 are not a common risk factor for SLE. It cannot be excluded that such variants might contribute to other heritable autoimmune diseases.

## 1. Introduction

SLE (Systemic Lupus Erythematosus) is a highly complex and heterogenous autoimmune disease with incompletely understood etiopathology [1,2]. Age of onset, specific organs affected, and the severity of the disease are highly variable between patients. SLE is characterized by a breakdown in immune tolerance, which promotes the formation of autoreactive B and T cells, abnormal cytokine production, and the subsequent generation of autoantibodies against DNA- and RNA-based self-antigens [2]. Women are nine times more frequently affected than men, and the incidence of the disease is highest in women of childbearing age [3].

SLE is thought to be caused by interactions between susceptibility genes and environmental factors resulting in an irreversible loss of immunologic self-tolerance. Several GWAS identified more than 100 risk loci for SLE, including associations to the HLA locus and many non-coding and presumably regulatory genome regions [4,5]. The X-chromosomal *TLR7* gene encoding toll-like receptor 7 is one of the confirmed risk loci for SLE [5]. Increased TLR7 activity promotes autoimmunity [6,7,8], and there are indications that partial escape of *TLR7* from X-chromosome inactivation may contribute to the extreme sex-bias in SLE incidence [9]. A single nucleotide variant in the 3′-UTR, rs3853839, modulates TLR7 expression and has been repeatedly associated with SLE [10,11].

While SLE is a genetically highly complex disease, rare patients exist in which SLE or related autoimmune disorders are caused by single-gene defects. Genes affected in such patients include *DNASE1* [12] and *TREX1* [13].

TLR7 is activated by single-stranded RNA and represents a component of the innate immune defense against RNA viruses. The trafficking chaperone UNC93B1 is required for the correct localization of TLR7 to endosomal membranes, and its loss-of-function leads to an immune deficiency [14,15,16]. TLR7 and UNC93B1 form a heterotetrameric complex in a 2:2 stoichiometry in endosomal membranes [17]. Subsequent to TLR7 activation, syndecan binding protein (SDCBP) binds to the C-terminal tail of UNC93B1, which induces the termination of TLR7 signaling [18]. Genetic variants in *UNC93B1* that prevent SDCBP binding without affecting other functions of UNC93B1 result in the disruption of this negative feedback loop and consequently overactive TLR7 signaling that will eventually be triggered by endogenous RNA molecules [18,19,20,21]. *Unc93b1^PKP/PKP^* mice with a targeted disruption of the SDCBP binding motif develop a fatal systemic inflammation and autoimmune disease [18]. In dogs, a spontaneous missense variant affecting the C-terminal tail of UNC93B1 causes exfoliative cutaneous lupus erythematosus (ECLE) [22]. ECLE starts with skin lesions but typically develops into a systemic form of lupus in the affected dogs [22,23,24,25,26] (Figure 1).

Based on the recent insights about UNC93B1 function and the phenotypes in *UNC93B1* mutant mice and dogs, we hypothesized that genetic variants affecting the C-terminal tail of UNC93B1 might also be responsible for SLE or related autoimmune disease in human patients. We therefore investigated the sequence of the last exon of the *UNC93B1* gene in patients of the Swiss SLE Cohort Study.

## 2. Materials and Methods

### 2.1. Patient Selection and DNA Extraction

This study included 536 patients of the Swiss SLE Cohort Study (SSCS). Of the patients, 457 (85%) were female and 414 (78%) of European descent. In 497 (93%) of patients, SLE was diagnosed after the age of 18 years. Details about this cohort have been reported previously [27,28,29]. Genomic DNA was isolated from ETDA blood samples with the Maxwell RSC Whole Blood Kit using a Maxwell RSC instrument (Promega, Dübendorf, Switzerland).

### 2.2. UNC93B1 Targeted Sanger Sequencing

A 1000 bp PCR amplicon was amplified with primers AAGGGACAGTGCTGGATGTG (Primer F) and CAGGGCATCCGTGCATCC (Primer R). This amplicon contained 198 bp of the last intron of *UNC93B1*, 312 bp protein-coding region of the last exon and 490 bp of the 3′-UTR. The protein-coding part corresponded to codons 495 to 597 of the open reading frame. PCRs were performed in 10 μL total volume containing 10 ng template DNA, 5 pmol of each primer, 5 μL AmpliTaqGold360Mastermix, and 1 μL of GC enhancer (Thermo Fisher Scientific, Waltham, MA, USA). A touchdown PCR was performed with an initial denaturation for 10 min at 95 °C, followed by 5 cycles of 30 s denaturation at 95 °C, 30 s annealing at 65 °C with a decrease of 1 °C at each cycle, and 60 s polymerization at 72 °C. Subsequently, 30 cycles of 30 s denaturation at 95 °C, 30 s annealing at 60 °C and 60 s polymerization at 72 °C followed. At the end, a final extension step of 7 min at 72 °C was performed. After treatment with shrimp alkaline phosphatase and exonuclease I, PCR amplicons were sequenced with the forward primer F and reverse primer R1 (AGCTGTGGGGATCTGGAGC) on an ABI 3730 DNA Analyzer (Thermo Fisher Scientific). Sequencher 5.1 software was used to analyze the Sanger sequences (GeneCodes, Ann Arbor, MI, USA).

### 2.3. Whole Genome Sequencing

An Illumina TruSeq PCR-free DNA library with ~400 bp insert size of patient no. 8 was prepared. We collected 373 million 2 × 150 bp paired-end reads or 33x coverage on a NovaSeq 6000 instrument. Variant calling was performed using a community-developed pipeline from bcbio nextgen v1.1.6a0 (https://github.com/bcbio/bcbio-nextgen, accessed 9 August 2021). In short, the reads were mapped to the human reference genome assembly GRCh38 using BWA-MEM v. 0.7.17 [30]. Variant calling was conducted using GATK HaplotypeCaller v. 3.8 [31] and FreeBayes v. 1.1.0.46 (https://github.com/freebayes/freebayes, accessed 9 August 2021) [32] and the union of quality-filtered calls from both tools was used for downstream analysis. The variants were annotated using Ensembl Variant Effect Predictor (VEP) v. 98.3 [33] using in-house scripts. The short-read alignments of *UNC39B1* were visually inspected for structural variants using the integrative genomics viewer (IGV) [34].

### 2.4. Gene Analysis

Numbering within the human *UNC93B1* gene corresponds to the NCBI RefSeq accessions NM_030930.4 (mRNA) and NP_112192.2 (protein).

## 3. Results

We successfully amplified and sequenced the last exon of the *UNC93B1* gene in all 536 investigated patients and identified six coding variants with respect to the reference sequence (Table 1, Appendix A).

Five variants did not affect the known SDCBP binding motif spanning amino acids 521 to 553 and were not investigated further. Only the p.Asp551Asn variant was located in the SDCBP binding domain of UNC93B1. In our cohort, one patient carried one copy of the mutant Asn-allele, which was also present once in the current gnomAD dataset of 128,930 alleles.

This female patient developed the first SLE manifestations including polyarthritis, acute cutaneous lupus, antinuclear, and anti-dsDNA antibodies at 16 years of age. In the following years, she developed alopecia, thrombocytopenia, and in particular, recurrent, difficult to treat acute and chronic skin lupus lesions. Additionally, at 30 years of age, she developed neuropsychiatric manifestations and chronic diffuse pain requiring intensive and chronic treatment complicated by drug allergies and recurrent viral infections. The patient was refractory to multiple SLE-targeted treatments. She died at the age of 44 years for reasons seemingly unrelated to SLE. We performed a whole-genome sequencing experiment on this heterozygous patient and evaluated the entire *UNC93B1* gene for the presence of potential additional loss-of-function variants. However, we did not detect anything unusual in the other parts of the gene and concluded that the patient had a fully functional second copy of the *UNC93B1* gene.

## 4. Discussion

In this study, we tested the hypothesis that genetic variants affecting the SDCBP binding domain of UNC93B1 might cause SLE. This hypothesis was developed from the observation that mouse and dog mutants with such variants exhibit severe autoimmune phenotypes [18,22]. The autoimmune phenotype in ECLE affected dogs with *UNC93B1* variants shows autosomal recessive inheritance [22]. Extrapolating from dogs, a hypothetical human patient with *UNC93B1*-related SLE would, therefore, be required to carry either bi-allelic *UNC93B1* variants abrogating SDCBP binding or a combination of one mutant allele encoding a UNC93B1 protein with impaired SDCBP binding together with a complete loss-of-function allele on the second chromosome. As we did not find such a genotype in our cohort of 536 analyzed SLE patients, we conclude that variants affecting the SDCBP binding domain of UNC93B1 do not represent a common genetic risk factor for SLE. The majority of the patients in the SSCS cohort are of European descent (78%). Hypothetical *UNC93B1* risk alleles might be more common in other populations.

Different from the current knowledge in dogs, the mode of inheritance in *Unc93b1* mutant mice is semi-dominant. Homozygous mutant *Unc93b1^PKP/PKP^* mice have a very severe autoimmune phenotype and die early as a consequence of their systemic autoimmune disease. Heterozygous *Unc93b1^WT/PKP^* mice exhibit only mild signs of autoimmune disease and enhanced TLR7 signaling [18]. Therefore, we cannot exclude the possibility that the heterozygous ^551^Asn allele observed in one of our patients contributed to her autoimmune disease, probably in combination with other risk factors.

It has to be cautioned that we focused our analysis on the published SDCBP binding domain of UNC93B1 (amino acids 521-553). To the best of our knowledge, the exact three-dimensional structure of the UNC93B1-SDCBP complex is currently unknown. There might be additional important amino acids in the C-terminal tail or other cytosolic parts of UNC93B1 that are also required for SDCBP binding. The gnomAD database and our study provide evidence that several very rare UNC93B1 alleles with amino acid substitutions in the C-terminal tail exist in the human population. Variants affecting other components of this regulatory pathway, for example, the enzymes mediating phosphorylation or ubiquitination of UNC93B1, might also contribute to autoimmune diseases.

Our results do not exclude the possibility that variants affecting the SDCBP binding domain of UNC93B1 might cause or contribute to other autoimmune phenotypes than SLE in human patients. The investigated patients mostly had adult-onset SLE [29]. *Unc93b1^PKP/PKP^* mice develop signs of systemic inflammation and antinuclear antibodies very early in life [18]. Homozygous *UNC93B1* mutant dogs with ECLE develop a cutaneous form of lupus as juveniles at a few months of age. In most affected dogs, this progresses to a systemic autoimmune disease with frequent involvement of the joints. However, the formation of antinuclear antibodies, a hallmark of SLE, is normally not seen in dogs with ECLE. These phenotypic differences between UNC93B1 mutant mice and dogs impede an accurate prediction of the resulting phenotype in human patients with homologous *UNC93B1* variants. We therefore think that further studies investigating the presence of genetic variants affecting the C-terminal tail of UNC93B1 in patient cohorts with other autoimmune phenotypes, including familial cases of extremely rare autoimmune disorders, are warranted.

## Figures and Tables

**Figure 1 genes-12-01268-f001:**
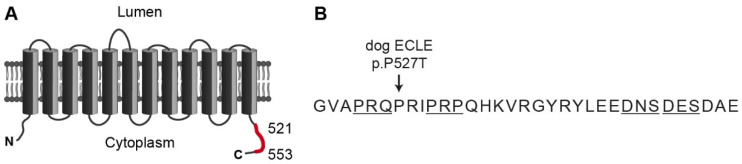
Topology of the human UNC93B1 protein. (**A**) UNC93B1 comprises 597 amino acids and contains 12 transmembrane domains. A segment of the C-terminal tail indicated in red is required for the interaction with SDCBP and subsequent dampening of TLR7 signaling [18]. (**B**) Amino acid sequence from position 521 to 553. Substitution of a highly conserved proline with threonine causes exfoliative cutaneous lupus erythematosus in dogs [22]. Targeted mutagenesis of the four underlined motifs disrupted SDCBP binding in mouse macrophages [18]. A targeted mouse mutant, *Unc93b1^PKP/PKP^*, in which the residues corresponding to the human positions 530–532 were replaced by alanines, developed systemic inflammation and autoimmunity [18].

**Table 1 genes-12-01268-t001:** Details of the six detected *UNC93B1* variants.

dbSNP	HGVS-c	HGVS-p	Alternative Allele Count (Frequency)	gnomAD Allele Frequency
rs7149	c.1557C>G	p.Arg519=	253 (23.6%)	16.0%
rs576491436	c.1629G>A	p.Glu543=	1 (0.1%)	5 × 10^−4^
rs1308430306	c.1651G>A	p.Asp551Asn	1 (0.1%)	8 × 10^−6^
n.a.	c.1724_1725delinsAG	p.Pro575Gln	8 (0.7%)	n.a.
rs2375182	c.1768G>T	p.Gly590Trp	4 (0.4%)	0.4%
rs964738111	c.1777G>A	p.Gly593Arg	1 (0.1%)	2 × 10^−4^

## Data Availability

The data presented in this study is contained within the article and Appendix A.

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
