# Peer review of "Variants Affecting the C-Terminal Tail of UNC93B1 Are Not a Common Risk Factor for Systemic Lupus Erythematosus"

_genes, 2021, doi:10.3390/genes12081268_

Round 1

Reviewer 1 Report

Dear Authors,

in this study Kiener et al. describe the screening of patients from the Swiss SLE Cohort Study for mutations in the C-terminal region of Unc93b1 to evaluate potential risk associations between SNPs in this region and disease. Unc93b1 is the trafficking chaperone of TLR7 required for receptor function. Specific mutations in the Unc93b1 c-terminus have previously been shown to induce autoimmune-like phenotypes in mice and dogs through enhancing TLR7 activity. The next logical step therefore is the search for such mutations in existing SLE patient cohorts, making the obtained data warranted and necessary. Although the authors could not identify risk-associated Unc93b1 mutations in the Swiss cohort, the results are still important and do not exclude the possibility that there might still exist Unc93b1-realted risk alleles associated with autoimmunity. The authors clearly lay out the limitations of their study and recommend to extent the search to other cohorts, with different autoimmune phenotypes or an earlier onset of disease.

The introduction and discussion are clearly explained and the results support the conclusions. Cited references are up to date and relevant to the study. There are a few minor points that are listed below, which would improve and complete the content. I would recommend accepting the article after minor revision.

Minor points:

The introduction provides a clear and concise entry into the field and logically builds up to the scientific question. For completeness, authors might want to include the human SNP rs3853839, located within the 3’ UTR of TLR7 that upregulates receptor expression and is a well-known risk-allele for lupus.

Typo line 96: the word ‘volume’ is written twice

I agree with the authors that these mutations are extremely rare, and don’t seem to be a common risk factor for SLE within the studied Swiss cohort.

In the discussion, authors might also want to mention the demographic limitation of their study (78% of European descent). The study cannot exclude the possibility that Unc93b1 risk alleles would be more commonly found in other demographics or maybe only affect single familial cases of extremely rare autoimmune disorders.

Author Response

(1)

The introduction provides a clear and concise entry into the field and logically builds up to the scientific question. For completeness, authors might want to include the human SNP rs3853839, located within the 3’ UTR of TLR7 that upregulates receptor expression and is a well-known risk-allele for lupus.

Response: We thank the reviewer for this comment. We added a sentence on rs3853839 and two additional references to the introduction.

(2)

Typo line 96: the word ‘volume’ is written twice

Response: Revised accordingly.

(3)

I agree with the authors that these mutations are extremely rare, and don’t seem to be a common risk factor for SLE within the studied Swiss cohort.

In the discussion, authors might also want to mention the demographic limitation of their study (78% of European descent). The study cannot exclude the possibility that Unc93b1 risk alleles would be more commonly found in other demographics or maybe only affect single familial cases of extremely rare autoimmune disorders.

Response: We fully agree with this comment. We slightly modified the discussion to better reflect these limitations of our study.

Reviewer 2 Report

This well-written manuscript analyzes UNC93B1 alleles in a cohort of SLE patients. Previous work implicated a region in the C-terminal tail of UNC93B1 as important for limiting TLR7 responses to self RNA. The authors find no evidence of an increased frequency of genetic variants in this region of UNC93B1 among the SLE patients. They conclude that variants in the C-terminal tail of UNC93B1 “do not represent a common genetic risk factor for SLE.” While this conclusion is supported by their data, I must say that I’m not surprised by the finding because it is already clear from existing databases that such coding variants could not be a common risk factor for SLE. The allele frequencies are extremely low. Nevertheless, I think it is nice that the authors have done this work and formally tested the hypothesis, so I am supportive of publication. I have three additional comments for how the text might be slightly modified:

  1. The authors make the point that the previously described UNC93B1 variants show autosomal recessive inheritance (lines 133-134). This may be true for the dog allele, but in our study the heterozygous UNC93B1-PKP mice did exhibit signs of SLE-like disease. The phenotype was milder than the homozygous mice, but disease was evident. The authors may want to revise their text to account for this fact.
  2. The authors allow for the possibility that variants in the C-terminal region of UNC93B1 may contribute to other autoimmune diseases, but they don’t discuss the possibility that mutations in other components of this regulatory pathway may also contribute to disease. The fact that phosphorylation and ubiquitylation of UNC93B1 appears to control this regulatory pathway suggests other way in which this process could become disrupted. Perhaps the authors feel commenting on this possibility would take them too far from their focus on UNC93B1, but they might consider mentioning this possibility in their discussion.
  3. Related to point #2, I feel the title might be more accurate if changed to “Genetic variants in the C-terminal tail of UNC93B1…”. The current title seems to exclude the possibility I raise in point #2, which I don’t think is the authors’ intention.

Author Response

(1)

The authors make the point that the previously described UNC93B1 variants show autosomal recessive inheritance (lines 133-134). This may be true for the dog allele, but in our study the heterozygous UNC93B1-PKP mice did exhibit signs of SLE-like disease. The phenotype was milder than the homozygous mice, but disease was evident. The authors may want to revise their text to account for this fact.

Response: We thank the reviewer for this very important comment. We revised the results and discussion to better reflect the semi-dominant inheritance in mice. As we have one heterozygous patient carrying a mutant UNC93B1Asn-551 allele in our cohort, in which this mutant allele might actually contribute to the phenotype, we added a very brief summary of the clinical history of this patient to the results.

(2)

The authors allow for the possibility that variants in the C-terminal region of UNC93B1 may contribute to other autoimmune diseases, but they don’t discuss the possibility that mutations in other components of this regulatory pathway may also contribute to disease. The fact that phosphorylation and ubiquitylation of UNC93B1 appears to control this regulatory pathway suggests other way in which this process could become disrupted. Perhaps the authors feel commenting on this possibility would take them too far from their focus on UNC93B1, but they might consider mentioning this possibility in their discussion.

Response: We added a sentence mentioning the possibility of functional variants in other components of the regulator pathway.

(3)

Related to point #2, I feel the title might be more accurate if changed to “Genetic variants in the C-terminal tail of UNC93B1…”. The current title seems to exclude the possibility I raise in point #2, which I don’t think is the authors’ intention

Response: We don’t intend to make any claims about variants in genes other than UNC93B1. According to our perception, the current title does not at all exclude the possibility of variants in other genes of the regulatory pathway. In our opinion, the alternative title proposed by the reviewer is not ideal as it combines “Genetic variant” with C-terminal tail, which is a part of the protein and not the gene. We therefore propose to keep the original title.